# Photosynthetic Response to Phosphorus Fertilization in Drought-Stressed Common Beech and Sessile Oak from Different Provenances

**DOI:** 10.3390/plants13162270

**Published:** 2024-08-15

**Authors:** Antonia Vukmirović, Željko Škvorc, Saša Bogdan, Daniel Krstonošić, Ida Katičić Bogdan, Tomislav Karažija, Marko Bačurin, Magdalena Brener, Krunoslav Sever

**Affiliations:** 1Faculty of Forestry and Wood Technology, University of Zagreb, Svetošimunska Cesta 23, HR-10000 Zagreb, Croatia; avukmiro@sumfak.unizg.hr (A.V.); zskvorc@sumfak.unizg.hr (Ž.Š.); sbogdan@sumfak.unizg.hr (S.B.); dkrstonosic@sumfak.unizg.hr (D.K.); ikaticic@sumfak.unizg.hr (I.K.B.); mbacurin@sumfak.unizg.hr (M.B.); mbrener@sumfak.unizg.hr (M.B.); 2Faculty of Agriculture, University of Zagreb, Svetošimunska Cesta 25, HR-10000 Zagreb, Croatia; tkarazija@agr.hr

**Keywords:** *Fagus sylvatica*, *Quercus petraea*, stomatal photosynthesis limitation, non-stomatal photosynthesis limitation, phosphorus nutrition, provenance-specific drought adaptation

## Abstract

Increasingly frequent and severe droughts pose significant threats to forest ecosystems, particularly affecting photosynthesis, a crucial physiological process for plant growth and biomass production. This study investigates the impact of phosphorus fertilization on the photosynthesis of common beech (*Fagus sylvatica* L.) and sessile oak (*Quercus petraea* (Matt.) Liebl.). In a common garden experiment, saplings originating from two provenances (wetter KA and drier SB provenances) were exposed to regular watering and drought in interaction with moderate and high phosphorus concentrations in the growing substrate. Results indicated that drought significantly reduced pre-dawn leaf water potential (Ψ_PD_), net photosynthesis (A_net_), stomatal conductance (g_s_) and photosynthetic performance index (PI_abs_) in both species. Phosphorus fertilization had a negative impact on A_net_ and PI_abs_, thus exacerbating the negative impact of drought on photosynthetic efficiency, potentially due to excessive phosphorus absorption by saplings. Provenance differences were notable, with the KA provenance showing better drought resilience. This research highlights the complexity of nutrient–drought interactions and underscores the need for cautious application of fertilization strategies in reforestation efforts under changing climatic conditions.

## 1. Introduction

Increasingly recurrent and severe droughts pose significant threats to forest ecosystems worldwide, making them a frequent subject of recent research [1,2,3,4,5,6]. Drought negatively affects the overall status of plants, and photosynthesis is one of the most important physiological processes inhibited by drought stress [7,8]. The negative impact on photosynthesis is reflected in plant vitality [9], biomass production [10], carbon storage [11] and the overall survival of natural forest ecosystems. During drought, both stomatal and non-stomatal limitations of photosynthesis occur. Drought conditions reduce water availability, leading to stomatal closure to minimize transpiration and avoid dehydration. This, in turn, limits CO_2_ intake and decreases photosynthetic efficiency [12,13]. Stomatal closure can also increase the production of reactive oxygen species (ROS) which are harmful to many organelles, especially the chloroplasts, where photosynthesis takes place [14,15]. Non-stomatal limitations are reflected in the diffusive resistance at the leaf level, affecting CO_2_ conductance in the mesophyll and chloroplasts, as well as causing metabolic impairments in the photosynthetic apparatus [7]. Stomatal limitations appear first when water is scarce, followed by non-stomatal limitations as the intensity and duration of water stress increase [16]. Severe drought can cause irreversible damage to the photosynthetic apparatus, preventing recovery after the drought subsides.

Due to their long lifespans, trees are unable to quickly adapt to environmental changes [17]. As drought and other disturbances hinder natural forest regeneration, cultivating plants in nurseries for reforestation becomes essential. Fertilizing saplings in nurseries is a common practice to enhance plant survival and growth [18]. However, the effectiveness of fertilization in drought-affected areas remains uncertain. Fertilization with complex mineral fertilizers typically increases total leaf area and reduces strict stomatal control [19]. Conversely, drought and other negative changes in European forest ecosystems worsen the mineral nutrition of forest trees, especially phosphorus (P) [20]. Phosphorus, second only to nitrogen, is a critical nutrient for plant productivity and its bioavailability in soil is strongly decreased by drought [21,22]. Phosphorus fertilization improves plant stress tolerance and has been shown to enhance photosynthetic capacity and growth under drought stress in many studies [15,23,24,25,26]. Its positive effect on plant growth during drought conditions has been linked to increased stomatal conductance, enhanced photosynthesis, improved cell membrane stability and better water relations [27].

When researching the response of forest trees to drought, special attention should be paid to their origin, i.e., provenance. Provenances may vary in their phenotypic plasticity for acclimation to drought and differences in the sensitivity of photosystem II to drought among provenances can also occur [28,29]. Acclimation to drought involves adjustments in several physiological traits, including photosynthesis, respiration, and changes in resource allocation [30]. Furthermore, trees and shrubs exhibit intraspecific genetic variation in traits related to drought resistance, including xylem vulnerability to cavitation, hydraulic conductance, water-use efficiency, stomata size and density and susceptibility to insect attack [31]. Stojnić et al. [17] found significant differences in embolism resistance among beech provenances, with those from southern Europe, which experience greater water deficits, being more resistant to embolism than those from northern Europe, where drought is less common.

In light of this, we investigated how phosphorus fertilization (at two different concentrations) affects the photosynthesis of common beech (*Fagus sylvatica* L.) and sessile oak (*Quercus petraea* (Matt.) Liebl.) saplings from two different provenances under drought conditions. Both species are ecologically and economically highly valuable in Europe, with beech being more drought-sensitive than oak [1,32,33,34,35]. Studies on phosphorus in beech have mainly examined inter- and intraprovenance differences in phosphorus uptake mechanisms, internal allocation during the growing season, leaf phenology and photosynthesis [36,37,38,39]. In oak, research has focused on the effect of phosphorus fertilization on total biomass production [40], growth [41] and leaf phenology [42]. Both species exhibit differences in drought acclimation between provenances [1,43]. Still, consensus is lacking on whether plants from drier or wetter provenances are better suited for the increasingly dry conditions expected in their natural distributions in the future [44].

The aims of this research were:to examine the effects of drought, phosphorus fertilization and provenance on the photosynthesis of common beech and sessile oak;to determine how phosphorus fertilization under drought conditions affects the stomatal and non-stomatal control of net photosynthetic CO_2_ assimilation in common beech and sessile oak;to examine the differentiation in photosynthetic performance in common beech and sessile oak provenances.

By examining the interplay between drought, phosphorus levels and provenance differences, the research addresses how phosphorus fertilization influences photosynthetic performance in these valuable European tree species. It aims to offer new insights that could improve forest management and reforestation strategies by identifying effective phosphorus fertilization practices.

## 2. Results

### 2.1. Soil Water Conditions

The volumetric water content (VWC) in regularly watered treatments was between 18.3 and 40.6% during the growing season. In the drought treatments, the VWC constantly decreased and at the peak of the drought was at 8.0 (in +PD treatment) and 10.1% (in −PD treatment).

Drought significantly affected the pre-dawn leaf water potential (Ψ_PD_) of both species (Figure 1, Figure 2A and Figure 3A). In regularly watered treatments (+PW and −PW) the Ψ_PD_ was not lower than −0.4 MPa throughout the growing season. In the drought treatments (+PD and −PD), the Ψ_PD_ decreased as the drought progressed, down to the lowest values at the peak of the drought (31 August) when it reached values of −3.1 MPa for beech and −3.3 MPa for oak (Figure 1).

### 2.2. Leaf Phosphorus Concentrations

Phosphorus fertilization significantly affected leaf phosphorus concentrations (P_leaves_) in both species (Figure 2B and Figure 3B). The range of mean P_leaves_ in common beech saplings fertilized with P (+PW and +PD treatments) was 1.79–1.97 mg P g^−1^ DW, indicating their nutrition with P between upper normal and surplus, while in the sessile oak saplings fertilized with P, it was 1.92–1.99 mg P g^−1^ DW, indicating their upper normal nutrition with P (according to the values given by Mellert and Göttlein [45]). The range of mean P_leaves_ concentration in common beech saplings non-fertilized with P (treatments −PW and −PD) was 1.49–1.69 mg P g^−1^ DW, indicating their mid-normal nutrition with P, while in the sessile oak saplings non-fertilized with P, it was 1.41–1.55 mg P g^−1^ DW, indicating their lower normal nutrition with P.

### 2.3. Sessile Oak Responses to Applied Effects

Drought significantly affected Ψ_PD_ (Figure 3A), A_net_ (Figure 3C), PI_abs_ (Figure 3D) and g_s_ (Figure 3E) in oak. These parameters were significantly lower in treatments exposed to drought (D) compared to regularly watered treatments (W). The effect of phosphorus was significant for P_leaves_, A_net_ and PI_abs_. Saplings in the +P treatment had significantly higher P concentration in leaves compared to saplings in the −P treatment (Figure 3B). Net photosynthesis (A_net_) (Figure 3C) and PI_abs_ (Figure 3D) were significantly lower in +P treatments compared to −P treatments. Provenances differed in A_net_ (Figure 3C) and PI_abs_ (Figure 3D). In both cases, higher values of A_net_ and PI_abs_ were found in saplings from the KA provenance compared to the SB provenance. The interaction effect of D × P × Pr was significant for A_net_ (Figure 3C) and PI_abs_ (Figure 3D). In both cases, the highest values of A_net_ and PI_abs_ were recorded in the −PW treatment, with no differences between provenances, and the lowest in the +PD treatment, also with no differences between provenances.

### 2.4. The Response of Photosynthetic Performance Parameters to Decreasing Leaf Water Potentials in Fertilized and Non-Fertilized Common Beech Saplings

Saplings from the −PD treatment from both provenances maintained higher values of A_net_ (Figure 4A) and PI_abs_ (Figure 4B) with a decrease in water potential compared to plants from the +PD treatment. The highest values were observed in saplings in the −PD treatment originating from the KA provenance. It is also evident that saplings from SB in the +PD treatment had the lowest PI_abs_ values (Figure 4B). Regarding g_s_, saplings from KA maintained the highest values with a decrease in water potential in both the −PD and +PD treatments, while saplings from SB had lower g_s_ values with a decrease in water potential, especially in the −PD treatment (Figure 4C). All correlations between Ψ_PD_ and A_net_, PI_abs_, as well as g_s_ are significant as indicated by *p* < 0.05 (Figure 4).

### 2.5. The Response of Photosynthetic Performance Parameters to Decreasing Leaf Water Potentials in Fertilized and Non-Fertilized Sessile Oak Saplings

With a decrease in water potential, the highest values of A_net_ and PI_abs_ were maintained by saplings from the KA provenance in the −PD treatment (Figure 5A,B). Regarding A_net_, the treatments were separated so that saplings from the −PD treatment had higher A_net_ with a decrease in water potential compared to saplings from the +PD treatment, regardless of provenance (Figure 5A). The same dynamic was recorded for g_s_ with a decrease in water potential (Figure 5C). All correlations between Ψ_PD_ and A_net_, PI_abs_, as well as g_s_ are significant as indicated by *p* < 0.05 (Figure 5).

## 3. Discussion

### 3.1. Effect of Drought on the Photosynthesis of Common Beech and Sessile Oak

Net photosynthesis was significantly lower in drought compared to regularly watered treatments in both beech and oak. This is a common response to drought stress: the rate of photosynthesis decreases due to either stomatal or non-stomatal limitations, as observed in previous studies on beech [46,47,48] and oak [33]. Drought stress triggers a series of physiological and functional adjustments in trees, with the most flexible responses, such as stomatal closure to reduce transpiration, occurring first as one of the earliest responses of plants to drought stress [8,49]. In both species, stomatal conductance was significantly lower in drought treatment compared to regularly watered treatment (Figure 2E and Figure 3E), indicating stomatal limitation. However, this negatively impacts photosynthesis because the inflow of CO_2_ decreases, resulting in stomatal limitation of photosynthesis [50]. Similar findings were reported in other studies [51,52]. Besides stomatal limitations, the rate of net photosynthesis can be further limited during drought due to non-stomatal disturbances. Photosystem II (PSII) impairment is often mentioned in this context, with previous research showing that PSII is damaged during drought, consequently negatively affecting net photosynthesis [53]. In our study, drought negatively affected PI_abs_ in both beech and oak (Figure 2D and Figure 3D); as PI_abs_ decreased, the net photosynthesis rate also decreased. Similar results were obtained by Luo et al. [8]. Thus, during drought stress, the rate of net photosynthesis in both species was lower in drought compared to regularly watered treatments due to both stomatal and non-stomatal disturbances. Similar findings were reported by Liu et al. [23] on bamboo, Chtouki et al. [54] on chickpeas and Iqbal et al. [52] on cotton. Pilon et al. [55] found similar results in their study on peanuts, indicating that although decreases in photosynthesis were closely linked to stomatal conductance, the main factors leading to reduced photosynthetic rates were primarily non-stomatal.

### 3.2. Effect of Phosphorus Fertilization on the Photosynthesis of Common Beech and Sessile Oak

In both species, phosphorus fertilization had a negative effect on PI_abs_ (Figure 2D and Figure 3D). Also, in both species, it is visible that the −PD treatments maintained higher values with a decrease in water potential compared to the +PD treatments (Figure 4B and Figure 5B). These results contradict the results of previous research on some tree and shrub species in which P application had a positive effect on PSII [23,24,25,56]. Although our plants did not have phosphorus concentrations in their leaves that would indicate a surplus, they were very close. Takagi et al. [57] emphasized that homeostatic functions are less effective under high inorganic phosphorus (Pi) conditions. Furthermore, they say that high Pi accumulation would cause an over-reduction of the photosynthetic electron transport chain. It is possible that in our research, plants fertilized with phosphorus accumulated too much phosphorus and manifested the symptoms of phosphorus toxicity. Phosphorus, like any other nutrient, can become toxic if accumulated by plants in high concentrations [57,58]. Thus, in the species *Pistacia vera* L., excessive phosphorus caused a toxic effect on net photosynthesis, stomatal conductance, chlorophyll fluorescence and chlorophyll content [58]. We assume a similar thing happened in our research with both species, and the reason for this may be that excessively high phosphorus concentrations negatively affected the balance with other nutrients, as in previous research where the negative effect of excessive P concentrations was manifested for this reason [58]. Furthermore, it has been demonstrated that high fertilization does not benefit drought-exposed oak seedlings in terms of the allocation of new assimilates [19]. Our research has shown that high fertilization does not benefit the photosynthesis of oak and beech, but quite the opposite.

It has been demonstrated that fertilization can mitigate the negative effects of drought on plant growth by promoting better regulation of water use efficiency and enhancing the activity of antioxidant enzymes [15]. Furthermore, previous research has shown that nutrient availability before drought has both positive and negative impacts on survival [59]. Phosphorus fertilization negatively affected the net photosynthesis rate in both species (Figure 3C and Figure 4C). These are the expected results considering the pronounced negative impact of P fertilization on PSII. Gessler et al. [59] stated that when a nutrient threshold is exceeded, higher nutrient availability predisposes plants to drought-induced mortality because of morphological and physiological traits that make trees less resistant to drought. Even with regularly watered plants, it had a negative impact, which is not in line with some previous research that pointed out that fertilization is highly effective in soils without water stress [21]. This indicates how too much of a certain nutrient causes nutritional imbalance and negatively affects the overall status of the plant.

### 3.3. Effect of Provenance and Drought × Phosphorus Fertilization × Provenance Interaction on the Photosynthesis of Common Beech and Sessile Oak

The research on the role of provenance in adaptation to drought has yielded diverse results. Some studies on oaks have shown a wide range of sensitivity to drought among provenances, while others have questioned whether provenances from drier habitats are indeed better adapted to drought than those from wetter habitats, which is generally assumed [1,34,60]. In our study, oaks from the KA provenance exhibited higher A_net_ and PI_abs_ values. The KA provenance had a higher average annual precipitation from 1949 to 2019 compared to the SB provenance, suggesting that plants from wetter provenances (KA) might be better adapted to drought than those from drier provenances (SB). However, during the period from 2016 to 2020 (the growth period of the investigated saplings in their natural habitats), the KA provenance experienced seventeen moderately to extremely dry months (nine during the growing seasons), while the SB provenance experienced nine moderately to extremely dry months (four during the growing seasons). Thus, during the saplings’ growth in their natural stands, Karlovac (KA) was a drier habitat than Slavonski Brod (SB), possibly explaining why the plants from the KA provenance were better adapted to drought in our research.

Hackett-Pain et al. [43] noted that beech provenances exhibit a high degree of adaptation to the local climate. Previous research has shown that phenotypic plasticity in drought resistance traits exists among different European beech provenances [61]. In our study, beeches from the KA provenance had higher PI_abs_ values, indicating they tolerated drought better than plants from the SB provenance. We attribute this to the climatic conditions during the saplings’ growth in natural stands, similar to the explanation for oaks. However, higher PI_abs_ values in saplings from KA did not significantly affect A_net_ in these saplings. This indicates that although there are differences in non-stomatal traits, these differences are not pronounced enough to affect overall net photosynthesis in our study. Overall, our results on beech and oak indicate differences between provenances that can affect physiological performance during drought stress. Further investigation is needed to understand these differences, and they should be considered when choosing genetic material or seedlings for natural forest stands.

The impact of drought, phosphorus fertilization and provenance interaction resulted in reduced A_net_ rates, likely due to impaired functioning of photosystem II (PSII) as indicated by lower PI_abs_ in oaks in regularly watered treatments. It is possible that P fertilization exacerbated the over-reduction of the photosynthetic electron transport chain, a phenomenon observed in previous research [57]. Since this was significant only in the regularly watered treatments and not in the drought treatments, it means that phosphorus did not significantly harm photosynthesis under drought conditions, but good water supply caused the saplings to absorb too much phosphorus. This also highlighted the difference among provenances, which means that provenances react differently to phosphorus fertilization only under well-watered conditions in our research.

## 4. Materials and Methods

### 4.1. Plant Material and Provenance Local Habitat Conditions

The research was conducted on four-year-old saplings of common beech and sessile oak from two mature mixed stands (provenances) dominated by these species in the continental part of Croatia. One provenance, 100 years old, is located in the north-western part near the city of Karlovac (KA provenance, 15.524041 E, 45.466135 N, 170 m a.s.l.). The other, 105 years old, is situated in the eastern part near the city of Slavonski Brod (SB provenance, 17.973173 E, 45.273451 N, 245 m a.s.l.) (Figure 6).

In early March 2021, saplings were carefully excavated with minimal damage to their root systems from under mature trees, at least 100 m apart in both provenances. More detailed habitat conditions of the investigated provenances, including phytosociological, geomorphological, climatological, meteorological and soil traits, are provided by Sever et al. [62].

Between 1949 and 2019, the mean annual precipitation in the KA provenance (1111.8 mm) was higher than in the SB provenance (770.3 mm). However, from 2016 to 2020, during the saplings’ growth period, KA provenance experienced seventeen moderately to extremely dry months (nine during the growing seasons) compared to SB provenance, which had nine such months (four during the growing seasons).

Most physical and chemical soil traits, including phosphorus concentration, were similar in both provenances. In the KA provenance, the phosphorus concentration was 0.50 ± 0.32 mg P_2_O_5_ per 100 g of soil and in the SB provenance, it was 0.64 ± 0.21 mg P_2_O_5_ per 100 g of soil, indicating low phosphorus concentration in soils at a depth of 0–30 cm in both provenances.

### 4.2. Experimental Design and Growth Conditions

We transported the excavated four-year-old saplings, averaging 36.6 ± 8.01 cm in height, to the garden of the Faculty of Forestry and Wood Technology, University of Zagreb (45.82065 N, 16.02303 E), where we set up a common garden experiment. The saplings were transplanted into four wooden boxes (155 × 275 × 80 cm, with a volume of 3.41 m^3^), each filled with 3800 L of Klasmann TS 3 substrate, which had a P_2_O_5_ concentration of 160 mg/L. To increase the P_2_O_5_ concentration to 300 mg/L, two of the four boxes received 1182 g of triple superphosphate (Triplex) fertilizer, containing 45% P_2_O_5_, resulting in high concentrations of easily accessible phosphorus. The other two boxes remained unfertilized, maintaining a P_2_O_5_ concentration of 160 mg/L, indicating moderate phosphorus levels. After fertilization, in each box 25 common beech and 25 sessile oak saplings originating from the KA provenance were transplanted, and the same number of common beech and sessile oak saplings from the SB provenance (100 saplings per box), arranged randomly at a spacing of 20 × 18 cm. In total, 400 saplings were planted. During the 2021 growing season, the saplings were exposed to natural meteorological conditions and regularly watered in summer to aid acclimatization and survival. In the 2022 growing season, all boxes were covered with a transparent PVC roof to prevent natural precipitation. Two boxes (one fertilized and one unfertilized with P) were manually watered with 40 L every four days. The other two boxes (one fertilized and one unfertilized with P) were subjected to drought conditions from 15 May 2022 to 1 September 2022, with minimal watering of 20 L per box only when wilting leaves appeared, indicating drought stress. This occurred three times: late July, early August and mid-August. Thus, during the 2022 growing season, the saplings were exposed to four treatments: regular watering with phosphorus fertilization (+PW treatment), regular watering without phosphorus fertilization (−PW treatment), drought with phosphorus fertilization (+PD treatment) and drought without phosphorus fertilization (−PD treatment).

### 4.3. Soil Water Content

Seasonal water dynamics in the substrate of all treatments were monitored using a data logger and sensors to measure volumetric water content (VWC) in the soil (Spectrum Technologies, Inc., Aurora, CO, USA). Four sensors were installed at a depth of 5–20 cm in each treatment.

### 4.4. Measurements of Net Photosynthesis, Stomatal Conductance, Chlorophyll Fluorescence and Leaf Water Potential

At the beginning of the season, in each of the four treatments, 10 beech saplings from the KA provenance and 10 beech saplings from the SB provenance were labeled, and the same was carried out for oak. Therefore, 40 plants were labeled in each treatment, and measurements were always conducted on these same plants.

Instantaneous rates of net photosynthesis (A_net_) and stomatal conductance (g_s_) were measured using a portable photosynthesis system (LCpro+, ADC BioScentific Ltd., Hoddesdon, UK) equipped with a broadleaf cuvette on one sun-exposed and healthy leaf per sapling. All A_net_ and g_s_ measurements were conducted between 10:00 a.m. and 3:00 p.m. (Central European summer time). The conditions inside the cuvette were kept constant at 400 ppm CO_2_, a photon flux density of 1000 μmol m^−2^ s^−1^ and an air temperature of 25 ± 2 °C. The cuvette with a leaf enclosed was equilibrated for 1 min before reading.

We analyzed fast fluorescence kinetics on one sun-exposed and healthy leaf per sapling once per sapling between 09:00 and 11:00 on dark-adapted leaves using a portable plant efficiency analyzer (Pocket PEA, Hansatech Instruments Ltd., Norfolk, UK). The photosynthetic performance index (PI_abs_) was calculated from the fluorescence kinetics.

On three out of the ten selected common beech and sessile oak saplings originating from KA and SB provenances per treatment, pre-dawn leaf water potential (Ψ_PD_) was measured using a portable pressure chamber (Model 600 Pressure Chamber Instrument, PMS Instrument Company, Albany, OR, USA). Measurements were performed approximately every two weeks during the growing period (from 16 May to 28 October).

A_net_, g_s_ and chlorophyll fluorescence measurements were performed on the same days as the Ψ_PD_ measurements.

### 4.5. Leaf Phosphorus Concentrations

At the beginning of September (shortly after the drought period was interrupted by re-watering), from the labeled plants on which the measurements were conducted, three composite leaf samples were taken for the analysis of the phosphorus concentration. Each sample consisted of a total of 10 randomly selected leaves from 10 plants per species provenance in a box. Leaves were dried, ground, homogenized and subjected to chemical analysis. The concentration of P was determined spectrophotometrically according to standardized international protocols [63].

### 4.6. Statistical Analysis

All statistical analyses were conducted using the SAS statistical 15.1 software package (SAS Institute Inc., Cary, NC, USA). The assumptions of residual normality and homogeneity of variances were assessed using the Shapiro–Wilk test and Levene’s test, implemented through the GLM and UNIVARIATE procedures in SAS. Residuals were plotted against fitted values to check for variance homogeneity, and the residual distribution was also examined. A factorial ANOVA was used to evaluate the fixed effects of drought, phosphorus fertilization, and provenance, as well as their interactions on leaf water potential, phosphorus concentrations in leaves and photosynthetic parameters for each species individually. In all cases, a Tukey post hoc test was conducted to determine the significance of differences (*p* < 0.05) between the levels of the studied effects.

## 5. Conclusions

This study examined the effects of drought stress, phosphorus fertilization and provenance on the net photosynthesis (A_net_) of beech and oak saplings. Drought stress significantly reduced A_net_ in both species due to stomatal and non-stomatal limitations, including impaired PSII function. Phosphorus fertilization exacerbated the effects of drought, primarily through non-stomatal limitations, as suggested by lower PI_abs_, which is linked to PSII efficiency. We assume that plants fertilized with phosphorus accumulated too much phosphorus, which can lead to an over-reduction of the photosynthetic electron transport chain. There are notable differences in photosynthetic performance among provenances of common beech and sessile oak. Oaks from the KA provenance, which experienced drier conditions during the saplings’ growth period, exhibited higher A_net_ and PI_abs_ than those from the SB provenance. Similarly, beeches from the KA provenance showed higher PI_abs_ values, indicating better drought tolerance, though this did not significantly affect A_net_. These findings highlight the complex interactions between environmental stressors and genetic background, emphasizing the need for careful provenance selection and nutrient management in forest management practices. Further research on forest tree species is essential to fully understand these dynamics.

## Figures and Tables

**Figure 1 plants-13-02270-f001:**
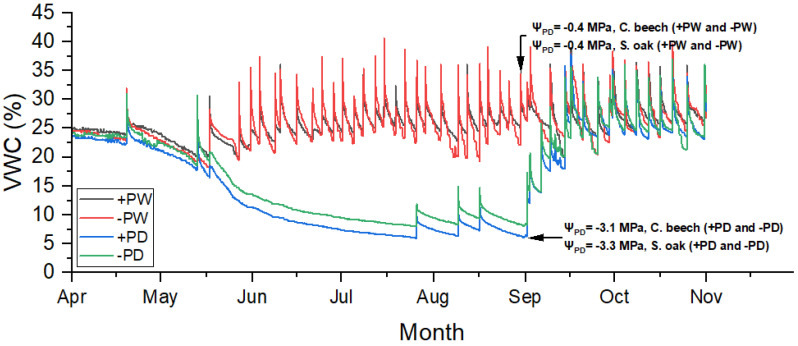
Patterns of substrate volumetric water content (VWC) in regularly watered and phosphorus fertilization treatment (+PW), regularly watered and non-phosphorus fertilization treatment (−PW), drought and phosphorus fertilization treatment (+PD) and drought and non-phosphorus fertilization treatment (−PD) with mean values of pre-down leaf water potential (Ψ_PD_) in regularly watered (+PW and −PW) and drought-treated (+PD and −PD) saplings of common beech (C. beech) and sessile oak (S. oak), measured on 26 May and 31 August.

**Figure 2 plants-13-02270-f002:**
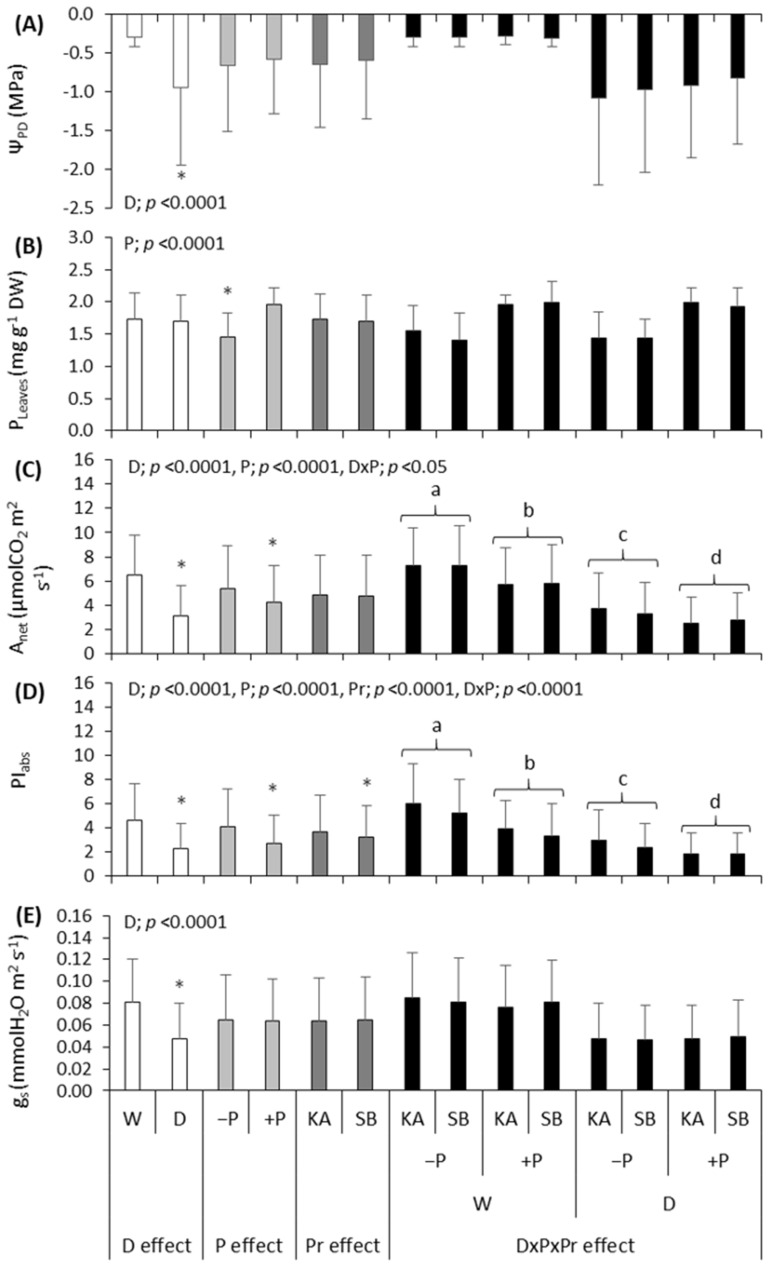
Mean season values ± SD of (**A**) leaf water potential (Ψ_PD_), (**B**) leaf phosphorus concentration (P_leaves_), (**C**) rate of net photosynthesis (A_net_), (**D**) photosynthetic performance index (PI_abs_) and (**E**) stomatal conductance (g_s_) for common beech; under the effect of drought (D effect; white bars) with regularly watered (W) and drought-treated (D) saplings, under the effect of phosphorus fertilization (P effect; light grey bars) with non-fertilized (−P) and fertilized (+P) saplings and under the effect of provenance (Pr effect; dark grey bars) with saplings originating from Karlovac (KA) and Slavonski Brod (SB) provenance, as well as with the effect of drought × phosphorus × provenance interaction (D × P × Pr effect; black bars), as calculated by factorial ANOVA. The pictures show *p* values which indicate the significance of the investigated effects. Asterix indicate significant differences between W and D saplings, −P and +P saplings, as well as between KA and SB provenance at *p* < 0.05. Lowercase letters indicate significant differences among saplings originating from KA and SB provenances depending on their water and fertilization treatments at *p* ˂ 0.05.

**Figure 3 plants-13-02270-f003:**
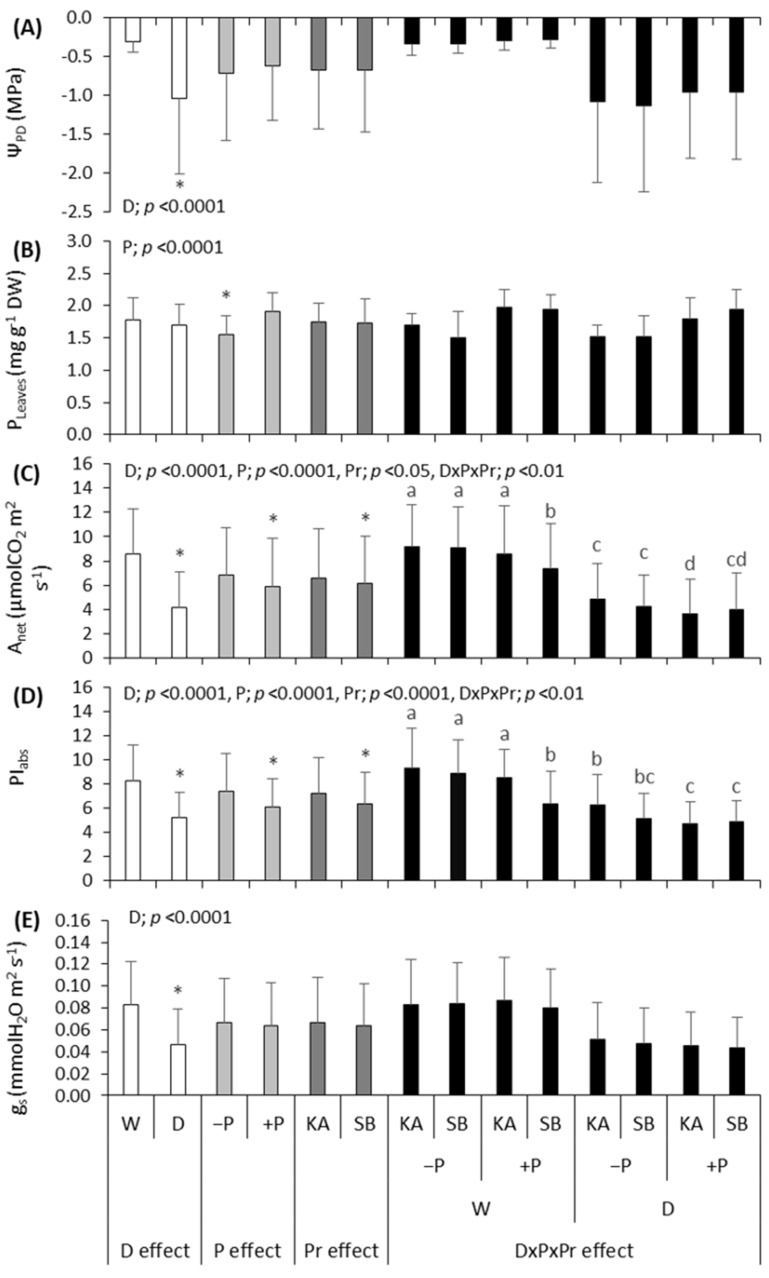
Mean season values ± SD of (**A**) leaf water potential (Ψ_PD_), (**B**) leaf phosphorus concentration (P_leaves_), (**C**) rate of net photosynthesis (A_net_), (**D**) photosynthetic performance index (PI_abs_) and (**E**) stomatal conductance (g_s_) for sessile oak; under the effect of drought (D effect; white bars) with regularly watered (W) and drought-treated (D) saplings, under the effect of phosphorus fertilization (P effect; light grey bars) with non-fertilized (−P) and fertilized (+P) saplings and under the effect of provenance (Pr effect; dark grey bars) with saplings originating from Karlovac (KA) and Slavonski Brod (SB) provenances, as well as with the effect of drought × phosphorus × provenance interaction (D × P × Pr effect; black bars), as calculated by factorial ANOVA. The pictures show *p* values which indicate the significance of the investigated effects. Asterix indicate significant differences between W and D saplings, −P and +P saplings, as well as between KA and SB provenance at *p* < 0.05. Lowercase letters indicate significant differences among saplings originating from KA and SB provenances depending on their water and fertilization treatments at *p* ˂ 0.05.

**Figure 4 plants-13-02270-f004:**
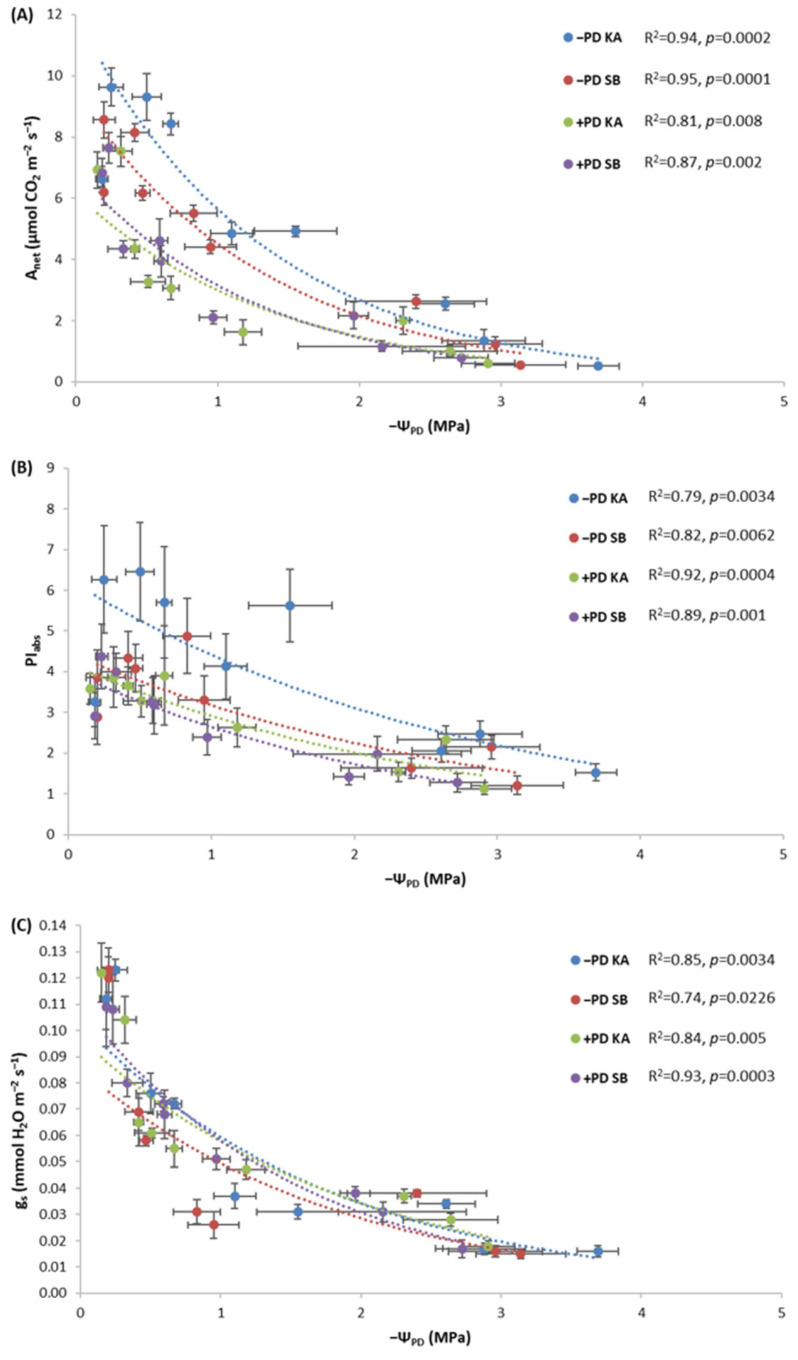
Dependence of (**A**) rate of net photosynthesis (A_net_), (**B**) photosynthetic performance index (PI_abs_) and (**C**) stomatal conductance (g_s_) on pre-dawn leaf water potential (−Ψ_PD_) in common beech saplings originating from Karlovac (KA) and Slavonski Brod (SB) provenances during the drought period, under fertilization with phosphorus and drought (+PD) and non-fertilization with phosphorus and drought (−PD) treatments. Results of the correlation analysis (R^2^ and *p*) are shown.

**Figure 5 plants-13-02270-f005:**
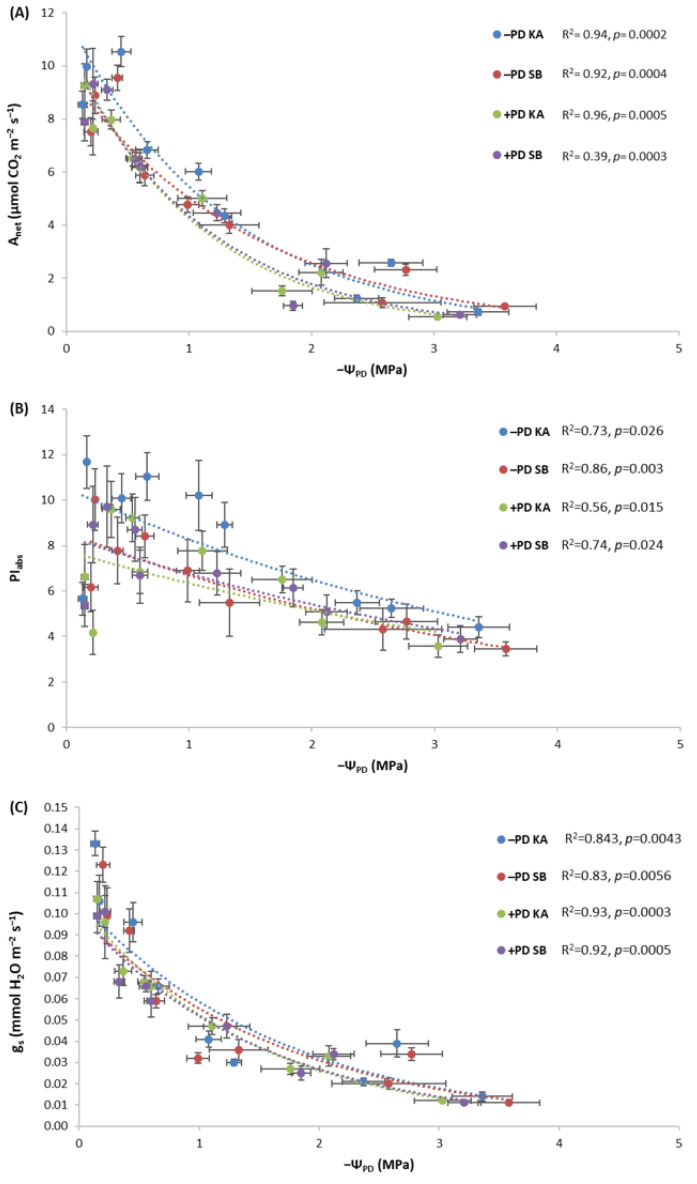
Dependence of (**A**) rate of net photosynthesis (A_net_), (**B**) photosynthetic performance index (PI_abs_) and (**C**) stomatal conductance (g_s_) on pre-dawn leaf water potential (−Ψ_PD_) in sessile oak saplings originating from Karlovac (KA) and Slavonski Brod (SB) provenances during the drought period, under fertilization with phosphorus and drought (+PD) and non-fertilization with phosphorus and drought (−PD) treatments. Results of the correlation analysis (R^2^ and *p*) are shown.

**Figure 6 plants-13-02270-f006:**
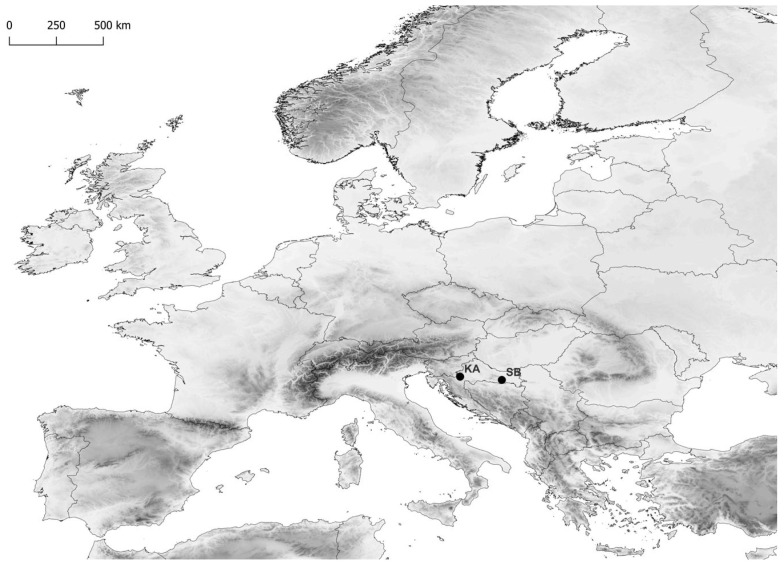
Geographical location of Slavonski Brod (SB) and Karlovac (KA) provenances.

## Data Availability

The data in this study are available from the corresponding author upon request.

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
