# Peer review of "Photosynthetic Response to Phosphorus Fertilization in Drought-Stressed Common Beech and Sessile Oak from Different Provenances"

_plants, 2024, doi:10.3390/plants13162270_

Round 1
Reviewer 1 Report
Comments and Suggestions for Authors
The author focused on drought significantly reduced net photosynthesis in common beech and sessile oak. This reduction was due to both stomatal and non-stomatal limitations, as indicated by lower gs and PIabs in drought compared to regularly watered treatments in both species.
This article is novel and within the journal's scope, but I would suggest some minor changes to the author, and I think it should be revised accordingly.
1. Please must improve the title.
2. The introduction is poorly written, must need to re-write, please avoid short paragraphs.
3. I suggest to improve the sub-headings of the results section.
4. Line 89, write as when it reached -3.1 MPa.
5. Line 99, the First word of the sentence must be a complete word.
6. I suggest adding a correlation analysis to further improve the quality of the result section.
Author Response
Dear Reviewer,
Please see the attachment.
Kind regards, Authors

Reviewer 2 Report
Comments and Suggestions for Authors
1. Please highlight the innovation of this study.
2. The introduction is too simple, please add some content.
3. The discussion is not deep enough. Please analyze the underlying reasons.
4. The references cited in the paper are too few, so more references in recent years should be cited.
5. The brand, model, origin and other information of the instrument used should be fully displayed.
6. What is the main question addressed by the research?
7. Are the conclusions consistent with the evidence and arguments presented? Do they address the main question posed?
8. Please highlight and emphasize the significance and purpose of this study.
9. The format of references is not standard, please modify it.
Author Response
Dear Reviewer,
please see the attachment.
Kind regards, Authors

Reviewer 3 Report
Comments and Suggestions for Authors
The manuscript on "Effects of drought, phosphorus fertilization and provenance on the photosynthesis of common beech and sessile oak" was prepared by nine authors from two institutions. This research highlights the complexity of nutrient-drought interactions and underscores the need for cautious application of fertilization strategies in reforestation efforts under changing climatic conditions.
The introduction concisely presents the relevance of the research and its stated goals. While the study encompasses extensive information, the introduction retains a notably general and brief nature. The research has three main objectives.
The results have been extensively presented, although the absence of highlighted materiality, particularly pertaining to top results and any significant changes, is notable.
The discussion addresses the topic, but the conclusions remain more in the realm of discussion rather than offering definitive outcomes. In the conclusion, it is crucial to present prominently the essential and significant results. The presentation should be concise, brief, and to the point.
Author Response

(The authors gave the same response as above.)

Round 2
Reviewer 2 Report
Comments and Suggestions for Authors
The paper has been strictly revised according to the reviewer's suggestions and is ready for acceptance.